# Task Arithmetic with LoRA for Continual Learning

**Rajas Chitale**[*], **Ankit Vaidya**[*], **Aditya Manish Kane**[*], **Archana Ghotkar**
Pune Institute of Computer Technology
{rajaschitale, ankitvaidya1905, adityakane1}@gmail.com,
aaghotkar@pict.edu

## Abstract

Continual learning refers to the problem where the training data is available in sequential chunks, termed "tasks". The majority of progress in continual learning has been stunted by the problem of catastrophic forgetting, which is caused by sequential training of the model on streams of data. Moreover, it becomes computationally expensive to sequentially train large models multiple times. To mitigate both of these problems at once, we propose a novel method to continually train transformer-based vision models using low-rank adaptation and task arithmetic. Our method completely bypasses the problem of catastrophic forgetting, as well as reducing the computational requirement for training models on each task. When aided with a small memory of 10 samples per class, our method achieves performance close to full-set finetuning. We present rigorous ablations to support the prowess of our method.

## 1 Introduction

The problem of continual learning has grown in importance in recent times due to the large-scale nature of downstream datasets. Moreover, the high costs of data labelling pose challenge to conventional fine-tuning, where all of the downstream data can be used at once. In many settings, it is necessary for an agent to learn on-the-fly, by either learning from experiences or by external supervision from humans. Continual learning has seen many applications, ranging from climate change (Kane et al., 2022), medical AI (Yi et al., 2023) to real-time chatbots (Liu and Mazumder, 2021). The most promising methods in continual learning maintain a small set of past samples, called "memory reservoir" (Chaudhry et al., 2019b), and try to avoid catastrophic forgetting by augmenting the batch of the current task with samples from the memory reservoir. This method has achieved success to some extent, but is far from perfect. This method still suffers catastrophic forgetting, and thus cannot be deemed as a reliable method for continual learning in general.

Vision Transformers (ViTs) (Dosovitskiy et al., 2021) have changed the way modern vision tasks are solved and have become ubiquitous in computer vision. Most of the prominent vision tasks have seen variants of ViTs reign supreme over the past couple of years. Efforts have been made to make ViTs, and transformers in general, run faster to support real-time applications. Low-Rank Adaptation (LoRA) (Hu et al., 2021) is a method for efficient fine-tuning where the large weight tensors are augmented by low-rank counterparts, which are trained along with the frozen weights of the original model.

Task arithmetic (Ilharco et al., 2023) is a simple method which leverages the semantics of weight spaces to manipulate the model weights to achieve compelling performance on tasks it was not trained

---

[*] first author, equal contribution
Code on GitHub

Workshop on Advancing Neural Network Training at 37th Conference on Neural Information Processing Systems (WANT@NeurIPS 2023).

on. It assumes that fine-tuning models pushes the weights away towards a semantically relevant direction in the weight space, thus creating a "task vector" from the pretrained checkpoint to the fine-tuned checkpoint.

We make a simple observation: modern continual learning demands the model to efficiently learn all tasks irrespective to the order they were provided. This observation points to a straightforward combination of task arithmetic, low-rank adaptation and memory reservoirs. In this paper, we show that this simple combination is indeed a powerful one. Specifically, we fine-tune only the low-rank weights of LoRA-ViT on each task. After training on individual tasks, we combine these low-rank weights using task arithmetic rules and merge them to the pre-trained ViT. Finally, we fine-tune this ViT on a small set of samples from the dataset. To evaluate our method, we perform experiments on Flowers-102 (Nilsback and Zisserman, 2008), Oxford-IIIT Pets (Parkhi et al., 2012) and CIFAR10 (Krizhevsky, 2009) datasets. We conclusively show that this simple procedure brings the ViT very close its fully trained counterparts, thus cementing the effectiveness of our method.

## 2    Related Work

In computer vision, CNN based models (He et al., 2015) have been the dominant architectures for tasks like classification, segmentation, and detection. After the application of the Transformer architecture (Vaswani et al., 2017) to these tasks in computer vision, Vision Transformers (ViTs) (Dosovitskiy et al., 2021) have surpassed the traditional models and have become the dominant paradigm. This architecture leverages the effectiveness of large-scale pre-training to surpass the previous state of the art performance in multiple vision tasks. A variant of this is the BEiT (Bao et al., 2021), which uses a similar pre-training method to BERT (Devlin et al., 2019) - masked image modelling. This model is then fine-tuned on downstream tasks, outperforming the ViT. There are also other variants like the DEiT (Touvron et al., 2021) which uses knowledge distillation to train Vision Transformers efficiently, ConViT (d'Ascoli et al., 2022) which introduces gated positional self-attention (GPSA) which can be equipped with a soft convolutional inductive bias for improved performance, UViT (Chen et al., 2022) which shows that ViTs can perform better on segmentation and detection tasks without the addition of CNN-like designs and the Swin Transformer (Liu et al., 2021) which introduces an architecture that can be used as a general-purpose backbone for vision tasks.

In continual learning, a sequence of contents like tasks or examples is provided incrementally over a period of time to the system, and it is expected that the system learns them as if they were provided simultaneously. One of the main hindrances in continual learning is catastrophic forgetting (French, 1999), where knowledge of previous tasks is lost when a system is trained on a new task. Some recent promising techniques to overcome catastrophic forgetting include as Experience Replay also called "memory reservoir" (Chaudhry et al., 2019b) in which a small set of past examples is maintained and the current task is augmented with these samples and AGEM (Chaudhry et al., 2019a) which stores the past examples and treats the losses on the past examples as an inequality constraint. Elastic Weight Consolidation (EWC) (Kirkpatrick et al., 2017) is another method used to mitigate catastrophic forgetting in which we selectively decrease the plasticity of weights and protects the previous knowledge while training on new tasks. Another method, Learning without Forgetting (Li and Hoiem, 2018) uses a combination of knowledge distillation and fine-tuning to preserve the performance on old tasks. Deep Generative Replay (Shin et al., 2017) is a method which uses fake data that is generated to mimic former examples in training to enable flexible knowledge transfer between tasks.

Fine-tuning models for downstream tasks is an inefficient process as all of the parameters are updated. One of the ways to overcome this inefficiency is to make use of parameter efficient fine-tuning, where we fine-tune only a small number of parameters with relatively unchanged performance. Some methods include Adapters (Houlsby et al., 2019) where we add a small number of parameters to the model which are trained for downstream tasks. However adapters introduce inference latency and bottlenecks which make the overall process more inefficient. Low-Rank Adaptation (LoRA) (Hu et al., 2021) uses low-rank matrix counterparts of the original weights during fine-tuning, and keeps the actual weights frozen. After training is done the model parameters are updated using the low-rank matrices at no inference cost or bottlenecks.

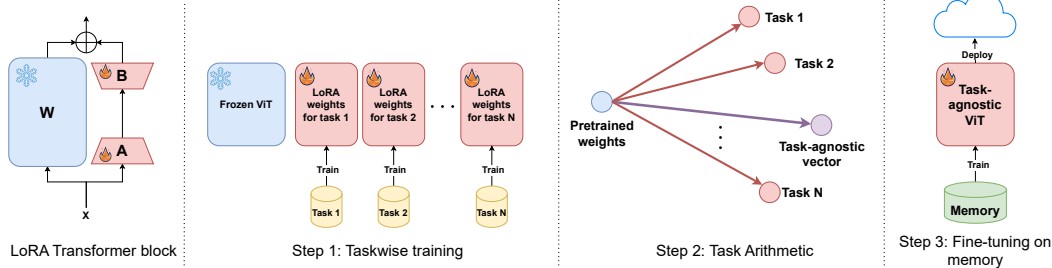

Figure 1: Our experimental setup. We only train the LoRA weights associated with the model. Each input example passes through the frozen and LoRA weights, and both the outputs are summed to get the final output. Each of these sets of LoRA weights is trained on the data of a particular task. We then calculate and merge the task vectors associated with each of these tasks. Finally, we fine-tune the model on a small set of samples, after which the model is ready for deployment.

## 3 Method

### 3.1 Problem setting

We address the problem of continual learning in a class-incremental setting. Concretely, there are $N$ tasks, each denoted by $T^i = \{X^i, Y^i\}$ where $i \in \{0, \ldots, N-1\}$ and $X^i$ and $Y^i$ are image-label pairs for the $i^{th}$ task. Here each element in $Y^0 \cup Y^1 \cup \ldots \cup Y^{N-1} \subset \{0, \ldots, C-1\}$, and $C$ is the number of total classes in the dataset. Note that $X^0 \cap X^1 \cap \ldots \cap X^{N-1} = \phi$, which means no samples are repeated across tasks. Each task encompasses a subset of classes from the total number of classes, which roughly equals $C/N$. A model is sequentially provided the data associated with each task $T^i$, with the preceding task's data being inaccessible when a new task is presented. The goal, in our experimental setup, is to maximize performance on a hold-out set $\{X^{test}, Y^{test}\}$ where elements from $Y^{test} \subset \{0, \ldots, C-1\}$. The most well-known problem in this setting is "catastrophic forgetting", where the model performs poorly on initial tasks, since the weights are overridden by the updates while training on newer tasks.

### 3.2 LoRA-adapted task-wise training

The first part of our approach is to take an off-the-shelf ViT backbone and augment it with LoRA. Specifically, we introduce an additional set of weights, corresponding to all query and value weight matrices in ViT. These new weights have a drastically low rank, and a single matrix $\underset{M \times M}{W}$ is represented by two matrices of lower rank, $\underset{M \times K}{A}$ and $\underset{K \times M}{B}$ where $K << M$. In practice, $M = 768$ and $K = 16$. So, for an input tensor $\underset{b \times s \times M}{X}$ ($b$ and $s$ are batch size and sequence length, respectively):

$$output = WX + BAX \tag{1}$$

The advantage of this method is that while training, $W$ is frozen, and only $A$ and $B$ are updated. This results in significantly lesser computation, both in terms of FLOP count and wall-clock time. For inference, we modify the model weights such that

$$W^* = (W + BA)X \tag{2}$$
$$output = W^*X \tag{3}$$

This effectively makes it the same as a conventional fine-tuned model. In our experiments, we fine-tune each LoRA-augmented ViT on a specific task. We finally merge the LoRA weights to the base model to get the task-specific model.

### 3.3 Task arithmetic

For all experiments, we assume the "task vector" of a given task-specific model to be defined as follows:

$$\tau_i = \theta_i - \theta_{pre} \tag{4}$$

Here, $\theta_i$ is the collection of weights of the task-specific model of task $T^i$, and "$-$" implies elementwise subtraction. After fine-tuning each LoRA-augmented model on its respective task, we combine the task vectors using the following:

$$\tau = \Sigma_{i=0}^{N-1} \lambda \tau_i \tag{5}$$

Intuitively, we get our final, task-agnostic model by doing the elementwise operation:

$$\theta_{final} = \theta_{pre} + \tau \tag{6}$$

This, in itself provides a surprisingly strong benchmark on two of our three evaluation datasets.

## 3.4 Finetuning on memory

Commonly used methods like episodic memory (Chaudhry et al., 2019b) and experience replay (Rolnick et al., 2019) use a memory buffer to capture examples from individual tasks and to retrain the model later on these examples. To emulate this effect, we choose 10 samples per class from each dataset and fine-tune our final, task-agnostic model on this collated set. This approach has two advantages over the experience replay approach. Firstly, there is no requirement of augmenting every batch as done in reservoir sampling, for example. Secondly, training the model on a balanced dataset completely bypasses the inconsistencies associated with sampling. This not only improves the performance, but brings it very close to the full-shot supervised baseline. We present our experimental setup and results in the upcoming section.

## 3.5 Enforcing feature distribution using KL-divergence loss

One of the intuitions behind training only LoRA weights is that this will result in small changes to the model, which will make more sense when we perform task arithmetic. Extending this thought, we also enforce the similarity of distribution from LoRA-adapted model to the one from vanilla ViT. This would enforce the task-wise LoRA features to be closer to the more generalized vanilla ViT features. Our loss function for KL-divergence experiments is as follows:

$$\mathcal{L}_{cls} = Crossentropy(outputs, Y) \tag{7}$$

$$\mathcal{L}_{KL} = KLDiv(softmax(f_{bb}^{pre}(x)), softmax(f_{bb}^{ft}(x)) \tag{8}$$

$$\mathcal{L} = \lambda_1 \mathcal{L}_{cls} + \lambda_2 \mathcal{L}_{KL} \tag{9}$$

Here, $f_{cls}$ and $f_{bb}$ are the classifier head and backbone respectively, where the superscript denotes if the model is pre-trained or fine-tuned. We take a weighted sum of the losses, and we empirically choose $\lambda_1 = 0.6$ and $\lambda_2 = 0.4$. We have illustrated our complete training procedure in Figure 1.

# 4 Results and discussion

The efficiency of our proposed method is demonstrated through varied experiments and comparisons. The experimental setup consists of three datasets namely, Oxford-IIIT Pets (37 classes), Flowers-102 (102 classes), and CIFAR10 (10 classes). Table 1 shows the details of the data samples used for training and testing. The task to class mapping for each dataset is shown in Appendix A.

## 4.1 Comparison with offline learning benchmarks

Pretrained ViT and its LoRA counterpart were trained in the offline setting on the three datasets to get the baseline accuracy for each dataset. We observed that the resultant model of our continual learning approach, when trained on Flowers-102, has results almost similar to the offline counterparts on the same dataset. Our method showed lower but comparable accuracy with the offline learning benchmarks when trained on Oxford-IIIT Pets and CIFAR10 datasets. These results can be viewed and compared from Table 2.

---

$\lambda$ in Eqn. 5 is not learnable and the experiments are performed with $\lambda = 0.25$.

Table 1: Details of the datasets used for experimentation

| Dataset | No. of tasks | Train | | Test | |
|---|---|---|---|---|---|
| | | **Total** | **Avg. samples/task** | **Total** | **Avg. samples/task** |
| Oxford-IIIT pets | 6 | 3680 | 613 | 3669 | 612 |
| Flowers-102 | 10 | 2040 | 204 | 6149 | 615 |
| CIFAR10 | 5 | 50000 | 10000 | 10000 | 2000 |

## 4.2 Comparison with continual learning benchmarks

In order to validate the efficiency of our proposed CL method, we benchmarked our results against the SOTA methods for CL like AGEM (Chaudhry et al., 2019a) and Experience Replay (ER) (Chaudhry et al., 2019b). The Table 2 shows the superior performance of the new approach as compared to the SOTA methods stated above. Moreover, as a consequence of training a LoRA-augmented ViT, we can observe a significant reduction in the training time and FLOPs.

Table 2: Top-1 accuracy(%) of ViT, LoRA-ViT trained on entire datasets, AGEM and Experience Replay (ER) trained by class-incremental CL and our proposed class-incremental CL approach, trained on different datasets (50 epochs). "XEnt" and "KLDiv" stand for Crossentropy and KL Divergence losses respectively. The best scores for the continual setting have been highlighted in **bold**.

| | | **Oxford-IIIT Pets** | **Flowers-102** | **CIFAR10** |
|---|---|---|---|---|
| **Offline learning baselines** | ViT | 94.03 | 98.69 | 99.05 |
| | LoRA ViT | 93.75 | 97.08 | 98.55 |
| **CL baselines** | AGEM | 73.18 (-20.85) | 33.39 (-65.3) | 62.3 (-36.35) |
| | Replay | 88.25 (-5.78) | 91.07 (-7.62) | 86.25 (-12.8) |
| **Proposed method** | XEnt loss | **90.32** (-3.71) | 94.36 (-4.33) | **95.59** (-3.46) |
| | XEnt + KLDiv loss | 88.69 (-5.34) | **97.06** (-1.63) | 92.49 (-6.56) |

## 4.3 Computation analysis

The comparison of computational complexity of our proposed method and Experience replay measured in PFLOPs (PetaFLOPs) has been shown in Table 3. Our method uses 3 - 5 times less PFLOPs while significantly outperforming the existing methods. We calculated the FLOPs required for training using the fvcore library. The FLOPs required for the forward and backward pass were considered. In our LoRA-adapted ViT the FLOPs required for the backward pass were significantly lower as compared to a ViT as in a LoRA-adapted model the model weights are frozen and only the LoRA weights get updated during the backward pass. This contributes greatly to the lower complexity of our proposed methodology. Furthermore, we also note that memory replay essentially uses double the compute, since the model is trained on a combination of task data as well as data sampled from the memory.

## 4.4 Implementation details

For performing extensive experiments on LoRA-augmented ViT, we had access to a Nvidia Tesla P100 GPU with 16GB HBM2 memory. We employed Adam optimizer with a learning rate of $5\epsilon - 6$, weight decay of $1\epsilon - 6$, and batch size of 32 for all the datasets. The LoRA parameters that were configured for ViT, using the PEFT library (Mangrulkar et al., 2022) from the Hugging Face API, were $r = 16$ and $alpha = 16$, where $r$ is the dimension used by update matrices and $alpha$ is the scaling factor. The bias parameters were set as non-trainable parameters. This configuration enabled us to train only 2.02% of the total 87.6M parameters of a ViT to get impressive results.

Table 3: Comparison of Computational Complexity (in PFLOPs) of proposed method with Experience Replay (ER). Reduction shows the how much the number of PFLOPs required on each dataset was reduced by using our method.

| Dataset | Replay | Our Method | Reduction |
|---|---|---|---|
| Oxford-IIIT Pets | 17.828 | 3.493 | 5.10× |
| Flowers-102 | 6.139 | 1.599 | 3.84× |
| CIFAR10 | 237.326 | 44.880 | 5.28× |

The experiments were performed by incorporating KL Divergence loss and few-shot finetuning by training Oxford-IIIT Pets for 50 epochs, CIFAR10 for 50 epochs, and Flowers-102 for 30 epochs. The scaling factor used for adding the task vectors is $\lambda = 0.25$ for all experiments.

We used Avalanche (Lomonaco et al., 2021) to calculate the CL baselines for AGEM and Experience Replay (ER) methods. For AGEM, 100 patterns per task were used and the memory size for ER was set to 200. The offline learning baselines were calculated using the above-given learning rate and weight-decay with 50 epochs for Oxford-IIIT pets and CIFAR10 while using 30 epochs for Flowers-102.

### 4.5 Observations

In this section, we present some key observations from our experiments.

1. **Oxford-IIIT Pets and CIFAR10 have better pre-finetuning results:** We observe that without finetuning, Flowers-102 has the worst performance. We speculate that this is because of the high variance and high number of classes in Flowers-102. We hypothesize that if the tasks have some underlying distributions, the task vector will be better directed, thus having a better task-agnostic model.

2. **KL Divergence loss reduces variance amongst task vectors:** We observe that KL Divergence loss was detrimental in the case of datasets with low number of classes. However, in the case of Flowers-102, it gave a significant boost. We speculate that this is because in the former case, KL Divergence loss was a too strong regularizer, which hindered actual learning. However, in the case of Flowers-102, it reduced the high variance in the resultant task vectors, hence exhibiting better performance.

3. **The efficacy of few-shot finetuning makes the case for task arithmetic:** As shown in Appendix B, we can see that finetuning greatly improves the performance of the model obtained using task-arithmetic. We consider this as a testimony to our original weight manipulation method. Since the model is able to achieve near offline results with very few samples, we deduce that the weights obtained by task-arithmetic are a good initialization for the model to learn task-agnostic representations.

## 5 Conclusion

In this work, we introduce a novel approach to tackle continual learning. We use task arithmetic and low-rank adaption to mitigate catastrophic forgetting. We empirically show that the combination of these three seemingly unrelated methodologies outperforms classical baselines. Since a considerable part of vision community has started working on ViTs, we believe this work can serve as a simple yet strong baseline for all future works in the field of continual perception.

A probable future work in this direction is to study the model combination logic in greater detail. Works like ZipIt! (Stoica et al., 2023) propose novel methods to manipulate the weights in order to support continual settings. Weight manipulation for continual and multi-task settings is still a nascent and little-understood field, which might provide greater insights and improvements to our method.

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

## Appendix

## A    Dataset Details

The Oxford-IIIT Pets dataset, that has 37 classes, was split into **6 disjoint tasks** as shown in Table 4. The Flowers-102 dataset, which has 102 classes, was split into **10 disjoint tasks** as shown in Table 5. The CIFAR10 dataset, which has 10 classes, was split into **5 disjoint tasks** as shown in Table 6.

## B    Task-wise results

The task-wise accuracy values obtained while performing experiments on each dataset are shown below in the following sections. Tables 7, 8, and 9 present the performance for Oxford-IIIT Pets, Flowers-102 and CIFAR10 datasets respectively.

Table 4: Task to class mapping in the CL setting for Oxford-IIIT Pets

| | | |
|---|---|---|
| Task 0 | american_bulldog, scottish_terrier, english_setter, newfoundland, Maine_Coon, British_Shorthair | 6 classes |
| Task 1 | Persian, boxer, english_cocker_spaniel, saint_bernard, Russian_Blue, Bombay | 6 classes |
| Task 2 | japanese_chin, Sphynx, german_shorthaired, basset_hound, samoyed, shiba_inu | 6 classes |
| Task 3 | staffordshire_bull_terrier, Siamese, wheaten_terrier, Abyssinian, keeshond, havanese | 6 classes |
| Task 4 | yorkshire_terrier, Bengal, great_pyrenees, Egyptian_Mau, pomeranian, beagle | 6 classes |
| Task 5 | american_pit_bull_terrier, Ragdoll, miniature_pinscher pug, Birman, leonberger, chihuahua | 7 classes |

Table 5: Task to class mapping in the CL setting for Flowers-102

| | | |
|---|---|---|
| Task 0 | alpine sea holly, buttercup, fire lily, anthurium, californian poppy, foxglove, artichoke, camellia, frangipani, azalea | 10 classes |
| Task 1 | canna lily, fritillary, ball moss, canterbury bells, garden phlox, yellow iris, balloon flower, cape flower, gaura, barbeton daisy | 10 classes |
| Task 2 | carnation, gazania, bearded iris, bird of paradise, cautleya spicata, germanium, bee balm, clematis, giant white arum lily, colt's foot | 10 classes |
| Task 3 | globe thistle, bishop of llandaff, great masterwort, globe flower, black-eyed susan, common dandelion, grape hyacinth, blackberry lily, corn poppy, columbine | 10 classes |
| Task 4 | blanket flower, cyclamen, hard-leaved pocket orchid, bolero deep blue, daffodil, hibiscus, bougainvillea, desert-rose, hippeastrum, bromelia | 10 classes |
| Task 5 | english marigold, japanese anemone, king protea, stemless gentian, lenten rose, petunia, sunflower, lotus, peruvian lily, pincushion flower | 10 classes |
| Task 6 | sweet pea, love in the mist, pink primrose, sweet william, magnolia, pink-yellow dahlia, sword lily, mallow, poinsettia, thorn apple | 10 classes |
| Task 7 | marigold, primula, tiger lily, mexican aster, prince of wales feathers, toad lily, mexican petunia, purple coneflower, tree mallow, monkshood | 10 classes |
| Task 8 | red ginger, tree poppy, moon orchid, trumpet creeper, rose, morning glory, ruby-lipped cattleya, wallflower, orange dahlia, siam tulip | 10 classes |
| Task 9 | water lily, osteospermum, silverbush, watercress, oxeye daisy, snapdragon, wild pansy, spring crocus, passion flower, spear thistle, windflower, pelargonium, | 12 classes |

Table 6: Task to class mapping in the CL setting for CIFAR10

| | | |
|---|---|---|
| Task 0 | airplane, automobile | 2 classes |
| Task 1 | bird, cat | 2 classes |
| Task 2 | deer, dog | 2 classes |
| Task 3 | frog, horse | 2 classes |
| Task 4 | ship, truck | 2 classes |

Table 7: Task-wise Top-1 Accuracy(%) on **Oxford-IIIT Pets** dataset for our proposed approach experimented on combination of Crossentropy loss, KL Divergence loss and memory fine-tuning . "XEnt" and "KLDiv" stand for Crossentropy and KL Divergence losses respectively. "TARV" and "MemFT" stand for task-agnostic resultant vector and few-shot fine-tuning on memory, respectively. The best scores for the continual setting have been highlighted in **bold**.

| Task | XEnt loss | | XEnt + KLDiv loss | |
|---|---|---|---|---|
| | **TARV** | **TARV+MemFT** | **TARV** | **TARV+MemFT** |
| 0 | 62.94 | 83.97 | 72.29 | 85.64 |
| 1 | 91.31 | 95.57 | 76.32 | 93.87 |
| 2 | 55 | 91.17 | 57.67 | 92.67 |
| 3 | 85.67 | 93.52 | 64.51 | 84.64 |
| 4 | 90.45 | 95.31 | 77.39 | 90.45 |
| 5 | 75.43 | 83.71 | 82.43 | 85.43 |
| Entire dataset | 76.8 | **90.54** | 71.77 | **88.78** |

Table 8: Task-wise Top-1 Accuracy(%) on **Flowers-102** dataset for our proposed approach experimented on combination of Crossentropy loss, KL Divergence loss and memory fine-tuning . "XEnt" and "KLDiv" stand for Crossentropy and KL Divergence losses respectively. "TARV" and "MemFT" stand for task-agnostic resultant vector and few-shot fine-tuning on memory, respectively. The best scores for the continual setting have been highlighted in **bold**.

| Task | XEnt loss | | XEnt + KLDiv loss | |
|---|---|---|---|---|
| | **TARV** | **TARV+MemFT** | **TARV** | **TARV+MemFT** |
| 0 | 27.64 | 92.86 | 24.53 | 98.14 |
| 1 | 27.95 | 86.61 | 25.17 | 95.15 |
| 2 | 7.83 | 93.21 | 14.6 | 99.22 |
| 3 | 21.45 | 98.21 | 17.96 | 97.86 |
| 4 | 32.32 | 92.13 | 28.87 | 93.92 |
| 5 | 41.49 | 97.72 | 39.09 | 96.76 |
| 6 | 49.48 | 98.45 | 36.79 | 98.45 |
| 7 | 9.09 | 94.46 | 5.09 | 97.64 |
| 8 | 49.72 | 97.22 | 30.81 | 98.99 |
| 9 | 29.50 | 90.65 | 25.75 | 95.54 |
| Entire dataset | 29.65 | **94.15** | 24.87 | **97.17** |

Table 9: Task-wise Top-1 Accuracy(%) on **CIFAR10** dataset for our proposed approach experimented on combination of Crossentropy loss, KL Divergence loss and memory fine-tuning . "XEnt" and "KLDiv" stand for Crossentropy and KL Divergence losses respectively. "TARV" and "MemFT" stand for task-agnostic resultant vector and few-shot fine-tuning on memory, respectively. The best scores for the continual setting have been highlighted in **bold**.

| Task | XEnt loss | | XEnt + KLDiv loss | |
|---|---|---|---|---|
| | **TARV** | **TARV+MemFT** | **TARV** | **TARV+MemFT** |
| 0 | 92.3 | 97.35 | 14.9 | 93.9 |
| 1 | 84.9 | 90.65 | 96.55 | 94.05 |
| 2 | 95.8 | 95.35 | 28.35 | 81.6 |
| 3 | 96.1 | 97.6 | 26.1 | 95.85 |
| 4 | 97.3 | 97 | 12.15 | 97.05 |
| Entire Dataset | 93.28 | **95.59** | 35.61 | **92.49** |

