# OpenReview forum: "Task Arithmetic with LoRA for Continual Learning"
_NeurIPS.cc/2023/Workshop/WANT — WANT@NeurIPS 2023 Poster_

### Official Review · Reviewer_WVxc · 2023-10-18
**The paper introduces a new approach to continually train the vision transformers (ViT) using low-rank adaptation and task arithmetic.  For each task, the system fine-tunes the low-rank  weights of LoRA-ViT, combines these low-rank weights using task arithmetic rules, merges these weights to the pre-trained ViT and then fine-tunes this ViT on a small set of samples from the dataset.  The proposed method overcomes the problem of forgetting and reduces the computational requirement for training models on each task. The experiments show that the proposed method outperforms two SOTA continual learning (CL) methods and achieves performance close to full-set fine-tuning on three datasets.**

**Rating:** 6
**Confidence:** 5

**Review:**

Pros:
The paper is well-written and thus easy to follow.
The proposed method is intuitive.
The results are clearly presented.

Cons:
The approach is of incremental novelty and contribution.
The ablation study only covers the effect of adding KL Divergence loss and fine-tuning on memory. More ablation experiments need to be done, including experiments on the number of samples per class for fine-tuning and comparison of different values of the $\lambda_1$ and $\lambda_2$ hyperparameters.
Tables 7, 8 and 9 show that task-agnostic resultant vector (TARV) without fine-tuning on memory works on Oxford-IIIT Pets but does not work on Flowers-102 and CIFAR10. There is not enough discussion of the efficiency of task arithmetic.
Table 2 provides comparison with two SOTA CL methods: AGEM and Replay. Both published in 2019, comparisons with newer methods need to be made.
The related work section has to be structured.

---

### Official Review · Reviewer_uL37 · 2023-10-25
**LoRA for Continual Learning is a simple and intuitive idea that seems to work well**

**Confidence:** 4

**Review:**

The paper proposes to use LoRA for class-incremental continual learning. Specifically, LoRA weights are trained for each task independently, then they are combined using task arithmetic and merged into the pre-trained ViT. After this, additional full-rank fine-tuning is performed using a small set (~10 per class) of examples to serve as experience replay.

**Pros:**

1. Intuitive method that looks practically applicable
2. The method is evaluated on 3 datasets
3. Experiments show comparable performance to offline learning while reducing computational requirements by a factor of 4-5
4. Selected baselines are relevant and sufficiently support the claim of the paper

**Cons:**

1. Experiments with rather simple datasets. Neither CIFAR10, nor Flowers-102 are challenging. I am personally not familiar with Oxford-IIIT pets.
2. The method is only computationally-compared to replay baseline. No comparison to AGEM.
3. Full-rank fine-tuning step is not ablated

**Formatting issues and typos:**

1. Line 48, missing space between "models(He". Please double-check all of your \cite commands, this is a common typo.
2. Equation formatting needs improvement. Personally, I think it looks below NeurIPS workshop standards. I would recommend either rewrite the equations in more standard way or to replace them with pseudo-pytorch code. Either would be a good option and a big improvement over the existing equations in my opinion and I don't have a preference.
3. Table formatting could be improved. It's standard to use \toprule and \bottomrule only for the top and bottom of the table and to add vertical lines between experiment setup and results (e.g., after the second column in Table 1 and Table 2.
4. Adding a footnote to Section 3.3 that lambda in Equaiton 5 is not learnable and the experiments were performed with lambda=0.25 (as specified in Section 4.4) would simplify reading.

---

### Official Review · Reviewer_8qLx · 2023-10-25
**Good paper, accept**

**Confidence:** 4

**Review:**

The paper "Task Arithmetic with LoRA for Continual Learning" presents a novel approach to address the challenges of continual learning, specifically the issues of catastrophic forgetting and computational expense. The authors propose a method that combines low-rank adaptation and task arithmetic to continually train transformer-based vision models. By doing so, they successfully bypass the problem of catastrophic forgetting and significantly reduce the computational requirements for training models on each task. Additionally, the authors demonstrate that their method, when supported by a small memory of 10 samples per class, achieves performance levels close to full-set fine-tuning.


I recommend accepting this paper for publication. It presents simple yet efficient solution to a catastrophic forgetting problem in continual learning, offering promising results. The paper is well-structured, conveying the motivation, methodology, and findings clearly. The experimental results and comparisons with existing approaches strengthen the validity of the findings. Overall, this paper makes a significant contribution to the field of continual learning and will be of interest to researchers and practitioners.

---

### Meta-Review · Area_Chair_T5Xf · 2023-10-27

**Recommendation:** Accept (Poster)
**Confidence:** 3

**Metareview:**

**Strengths:**
* Most reviewers found the paper to be well-written and easy to follow.
* Strong evaluation across multiple datasets and baselines along with relevant ablation studies. Promising experimental results comparable to offline learning while reducing compute by 4-5x.

**Weaknesses:**
* Evaluation datasets may be too simple.
* Missing ablation studies. For instance, full-rank fine-tuning, number of samples per class for fine-tuning, etc.
* At least one reviewer feels that the contribution is incremental and that the paper has limited novelty.
* Comparison to more recent work suggested. Both in the evaluation and related work sections.

This paper appears to be borderline, but most reviews appear to be leaning positive. I will thus give this an accept (poster).

---

### Decision · Program_Chairs · 2023-10-28

**Decision:**

Accept (Poster)

**Comment:**

We thank the authors for their time and contribution to WANT and we are pleased to share that after the reviewing process the paper has been accepted. Congratulations! We encourage the authors to consider reviewers' feedback for the improvement of the camera-ready version. We hope to see you in person at the workshop and brainstorm on efficient training research together!